# Stochastic Convex Optimization with Multiple Objectives

**Mehrdad Mahdavi**
Michigan State University
mahdavim@cse.msu.edu

**Tianbao Yang**
NEC Labs America, Inc
tyang@nec-labs.com

**Rong Jin**
Michigan State University
rongjin@cse.msu.edu

## Abstract

In this paper, we are interested in the development of efficient algorithms for convex optimization problems in the simultaneous presence of multiple objectives and stochasticity in the first-order information. We cast the stochastic multiple objective optimization problem into a constrained optimization problem by choosing one function as the objective and try to bound other objectives by appropriate thresholds. We first examine a two stages exploration-exploitation based algorithm which first approximates the stochastic objectives by sampling and then solves a constrained stochastic optimization problem by projected gradient method. This method attains a suboptimal convergence rate even under strong assumption on the objectives. Our second approach is an efficient primal-dual stochastic algorithm. It leverages on the theory of Lagrangian method in constrained optimization and attains the optimal convergence rate of $O(1/\sqrt{T})$ in high probability for general Lipschitz continuous objectives.

## 1   Introduction

Although both stochastic optimization [17, 4, 18, 10, 26, 20, 22] and multiple objective optimization [9] are well studied subjects in Operational Research and Machine Learning [11, 12, 24], much less is developed for  stochastic multiple objective optimization, which is the focus of this work. Unlike multiple objective optimization where we have access to the complete objective functions, in stochastic multiple objective optimization, only stochastic samples of objective functions are available for optimization. Compared to the standard setup of stochastic optimization, the fundamental challenge of stochastic multiple objective optimization is how to make appropriate tradeoff between different objectives given that we only have access to stochastic oracles for different objectives. In particular, an algorithm for this setting has to ponder conflicting objective functions and accommodate the uncertainty in the objectives.

A simple approach toward stochastic multiple objective optimization is to linearly combine multiple objectives with a fixed weight assigned to each objective. It converts stochastic multiple objective optimization into a standard stochastic optimization problem, and is guaranteed to produce Pareto efficient solutions. The main difficulty with this approach is how to decide an appropriate weight for each objective, which is particularly challenging when the complete objective functions are unavailable. In this work, we consider an alternative formulation that casts multiple objective optimization into a constrained optimization problem. More specifically, we choose one of the objectives as the target to be optimized, and use the rest of the objectives as constraints in order to ensure that each of these objectives is below a specified level. Our assumption is that although full objective functions are unknown, their desirable levels can be provied due to the prior knowledge of the domain. Below, we provide a few examples that demonstrate the application of stochastic multiple objective optimization in the form of stochastic constrained optimization.

**Robust Investment.** Let $\mathbf{r} \in \mathbb{R}^n$ denote random returns of the $n$ risky assets, and $\mathbf{w} \in \mathcal{W} \equiv \{\mathbf{w} \in \mathbb{R}_+^n : \sum_i^n w_i = 1\}$ denote the distribution of an investor's wealth over all assets. The return for an investment distribution is defined as $\langle \mathbf{w}, \mathbf{r} \rangle$. The investor needs to consider conflicting objectives such as rate of return, liquidity and risk in maximizing his wealth [2]. Suppose that $\mathbf{r}$ has a *unknown* probability distribution with mean vector $\boldsymbol{\mu}$ and covariance matrix $\Sigma$. Then the target

of the investor is to choose an optimal portfolio $\mathbf{w}$ that lies on the mean-risk efficient frontier. In mean-variance theory [15], which trades off between the expected return (mean) and risk (variance) of a portfolio, one is interested in minimizing the variance subject to budget constraints which leads to a formulation like:

$$\min_{\mathbf{w} \in \mathbb{R}^n_+, \sum_i^n w_i = 1} \left[ \left\langle \mathbf{w}, \mathrm{E}[\mathbf{rr}^\top] \mathbf{w} \right\rangle \right] \quad \text{subject to} \quad \mathrm{E}[\langle \mathbf{r}, \mathbf{w} \rangle] \geq \gamma.$$

**Neyman-Pearson Classification.** In the Neyman-Pearson (NP) classification paradigm (see e.g, [19]), the goal is to learn a classifier from labeled training data such that the probability of a false negative is minimized while the probability of a false positive is below a user-specified level $\gamma \in (0, 1)$. Let hypothesis class be a parametrized convex set $\mathcal{W} = \{\mathbf{w} \mapsto \langle \mathbf{w}, \mathbf{x} \rangle : \mathbf{w} \in \mathbb{R}^d, \|\mathbf{w}\| \leq R\}$ and for all $(\mathbf{x}, y) \in \Xi \equiv \mathbb{R}^d \times \{-1, +1\}$ the loss function $\ell : \mathcal{W} \times \Xi \mapsto \mathbb{R}_+$ be a non-negative convex function. While the goal of classical binary classification problem is to minimize the risk as $\min_{\mathbf{w} \in \mathcal{W}} [\mathcal{L}(\mathbf{w}) = \mathrm{E}[\ell(\mathbf{w}; (\mathbf{x}, y))]]$, the Neyman-Pearson targets on

$$\min_{\mathbf{w} \in \mathcal{W}} \mathcal{L}^+(\mathbf{w}) \quad \text{subject to} \quad \mathcal{L}^-(\mathbf{w}) \leq \gamma,$$

where $\mathcal{L}^+(\mathbf{w}) = \mathrm{E}[\ell(\mathbf{w}; (\mathbf{x}, y))|y = +1]$ and $\mathcal{L}^-(\mathbf{w}) = \mathrm{E}[\ell(\mathbf{w}; (\mathbf{x}, y))|y = -1]$.

**Linear Optimization with Stochastic Constraints.** In many applications in economics, most notably in welfare and utility theory, and management parameters are known only stochastically and it is unreasonable to assume that the objective functions and the solution domain are deterministically fixed. These situations involve the challenging task of pondering both conflicting goals and random data concerning the uncertain parameters of the problem. Mathematically, the goal in multi-objective linear programming with stochastic information is to solve:

$$\min_{\mathbf{w}} [\langle \mathbf{c}_1(\xi), \mathbf{w} \rangle, \cdots, \langle \mathbf{c}_K(\xi), \mathbf{w} \rangle] \quad \text{subject to} \quad \mathbf{w} \in \mathcal{W} = \{\mathbf{w} \in \mathbb{R}^d_+ : A(\xi)\mathbf{w} \leq \mathbf{b}(\xi)\},$$

where $\xi$ is the randomness in the parameters, $c_i, i \in [K]$ are the objective functions, and $A$ and $\mathbf{b}$ formulate the stochastic constraints on the solution where randomness is captured by $\xi$.

In this paper, we first examine two methods that try to eliminate the multi-objective aspect or the stochastic nature of stochastic multiple objective optimization and reduce the problem to a standard convex optimization problem. We show that both methods fail to tackle the problem of stochastic multiple objective optimization in general and require strong assumptions on the stochastic objectives, which limits their applications to real world problems. Having discussed these negative results, we propose an algorithm that can solve the problem optimally and efficiently. We achieve this by an efficient primal-dual stochastic gradient descent method that is able to attain an $O(1/\sqrt{T})$ convergence rate for all the objectives under the standard assumption of the Lipschitz continuity of objectives which is known to be optimal (see for instance [3]). We note that there is a flurry of research on heuristics-based methods to address the multi-objective stochastic optimization problem (see e.g., [8] and [1] for a recent survey on existing methods). However, in contrast to this study, most of these approaches do not have theoretical guarantees.

Finally, we would like to distinguish our work from robust optimization [5] and online learning with long term constraint [13]. Robust optimization was designed to deal with uncertainty within the optimization systems. Although it provides a principled framework for dealing with stochastic constraints, it often ends up with non-convex optimization problems that are not computationally tractable. Online learning with long term constraint generalizes online learning. Instead of requiring the constraints to be satisfied by every solution generated by online learning, it allows the constraints to be satisfied by the entire sequence of solutions. However, unlike stochastic multiple objective optimization, in online learning with long term constraints, constraint functions are *fixed* and *known* before the start of online learning.

**Outline.** The remainder of the paper is organized as follows. In Section 2 we establish the necessary notation and introduce the problem under consideration. Section 3 introduces the problem reduction methods and elaborates their disadvantages. Section 4 presents our efficient primal-dual stochastic optimization algorithm. Finally, we conclude the paper with open questions in Section 5.

## 2 Preliminaries

**Notation** Throughout this paper, we use the following notation. We use bold-face letters to denote vectors. We denote the inner product between two vectors $\mathbf{w}, \mathbf{w}' \in \mathcal{W}$ by $\langle \mathbf{w}, \mathbf{w}' \rangle$ where $\mathcal{W} \subseteq \mathbb{R}^d$ is a compact closed domain. For $m \in \mathbb{N}$, we denote by $[m]$ the set $\{1, 2, \cdots, m\}$. We only consider the $\ell_2$ norm throughout the paper. The ball with radius $R$ is denoted by $\mathcal{B} = \{\mathbf{w} \in \mathbb{R}^d : \|\mathbf{w}\| \leq R\}$.

**Statement of the Problem** In this work, we generalize online stochastic convex optimization to the case of multiple objectives. In particular, at each iteration, the learner is asked to present a

solution $\mathbf{w}_t$, which will be evaluated by multiple loss functions $f_t^0(\mathbf{w}), f_t^1(\mathbf{w}), \ldots, f_t^m(\mathbf{w})$. A fundamental difference between single- and multi-objective optimization is that for the latter it is not obvious how to evaluate the optimization quality. Since it is impossible to simultaneously minimize multiple loss functions and in order to avoid complications caused by handling more than one objective, we choose one function as the objective and try to bound other objectives by appropriate thresholds. Specifically, the goal of OCO with multiple objectives becomes to minimize $\sum_{t=1}^{T} f_t^0(\mathbf{w}_t)$ and at the same time keep the other objective functions below a given threshold, i.e.

$$\frac{1}{T} \sum_{t=1}^{T} f_t^i(\mathbf{w}_t) \leq \gamma_i, \; i \in [m],$$

where $\mathbf{w}_1, \ldots, \mathbf{w}_T$ are the solutions generated by the online learner and $\gamma_i$ specifies the level of loss that is acceptable to the $i$th objective function. Since the general setup (i.e., full adversarial setup) is challenging for online convex optimization even with two objectives [14], in this work, we consider a simple scenario where all the loss functions $f_t^i(\mathbf{w}), i \in [m]$ are i.i.d samples from an unknown distribution [21]. We also note that our goal is NOT to find a Pareto efficient solution (a solution is Pareto efficient if it is not dominated by any solution in the decision space). Instead, we aim to find a solution that (i) optimizes one selected objective, and (ii) satisfies all the other objectives with respect to the specified thresholds.

We denote by $\bar{f}^i(\mathbf{w}) = \mathrm{E}_t[f_t^i(\mathbf{w})], i = 0, 1, \ldots, m$ the expected loss function of sampled function $f_t^i(\mathbf{w})$. In stochastic multiple objective optimization, we assume that we do not have direct access to the expected loss functions and the only information available to the solver is through a stochastic oracle that returns a stochastic realization of the expected loss function at each call. We assume that there exists a solution $\mathbf{w}$ strictly satisfying all the constraints, i.e. $\bar{f}^i(\mathbf{w}) < \gamma_i, i \in [m]$. We denote by $\mathbf{w}_*$ the optimal solution to multiple objective optimization, i.e.,

$$\mathbf{w}_* = \arg\min \left\{ \bar{f}^0(\mathbf{w}) : \bar{f}^i(\mathbf{w}) \leq \gamma_i, i \in [m] \right\}. \tag{1}$$

Our goal is to efficiently compute a solution $\widehat{\mathbf{w}}_T$ after $T$ trials that (i) obeys all the constraints, i.e. $\bar{f}^i(\widehat{\mathbf{w}}_T) \leq \gamma_i, i \in [m]$ and (ii) minimizes the objective $\bar{f}^0$ with respect to the optimal solution $\mathbf{w}_*$, i.e. $\bar{f}^0(\widehat{\mathbf{w}}_T) - \bar{f}^0(\mathbf{w}_*)$. For the convenience of discussion, we refer to $f_t^0(\mathbf{w})$ and $\bar{f}^0(\mathbf{w})$ as the objective function, and to $f_t^i(\mathbf{w})$ and $\bar{f}^i(\mathbf{w})$ for all $i \in [m]$ as the constraint functions.

Before discussing the algorithms, we first mention a few assumptions made in our analysis. We assume that the optimal solution $\mathbf{w}_*$ belongs to $\mathcal{B}$. We also make the standard assumption that all the loss functions, including both the objective function and constraint functions, are Lipschitz continuous, i.e., $|f_t^i(\mathbf{w}) - f_t^i(\mathbf{w}')| \leq L\|\mathbf{w} - \mathbf{w}'\|$ for any $\mathbf{w}, \mathbf{w}' \in \mathcal{B}$.

## 3 Problem Reduction and its Limitations

Here we examine two algorithms to cope with the complexity of stochastic optimization with multiple objectives and discuss some negative results which motivate the primal-dual algorithm presented in Section 4. The first method transforms a stochastic multi-objective problem into a stochastic single-objective optimization problem and then solves the latter problem by any stochastic programming approach. Alternatively, one can eliminate the randomness of the problem by estimating the stochastic objectives and transform the problem into a deterministic multi-objective problem.

### 3.1 Linear Scalarization with Stochastic Optimization

A simple approach to solve stochastic optimization problem with multiple objectives is to eliminate the multi-objective aspect of the problem by aggregating the $m+1$ objectives into a single objective $\sum_{i=0}^{m} \alpha_i f_t^i(\mathbf{w}_t)$, where $\alpha_i, i \in \{0, 1, \cdots, m\}$ is the weight of $i$th objective, and then solving the resulting single objective stochastic problem by stochastic optimization methods. This approach is in general known as the weighted-sum or scalarization method [1]. Although this naive idea considerably facilitates the computational challenge of the problem, unfortunately, it is difficult to decide the weight for each objective, such that the specified levels for different objectives are obeyed. Beyond the hardness of optimally determining the weight of individual functions, it is also unclear how to bound the sub-optimality of final solution for individual objective functions.

### 3.2 Projected Gradient Descent with Estimated Objective Functions

The main challenge of the proposed problem is that the expected constraint functions $\bar{f}^i(\mathbf{w})$ are not given. Instead, only a sampled function is provided at each trial $t$. Our naive approach is to replace the expected constraint function $\bar{f}^i(\mathbf{w})$ with its empirical estimation based on sampled objective functions. This approach circumvents the problem of stochastically optimizing multiple objective

into the original online convex optimization with complex projections, and therefore can be solved by projected gradient descent. More specifically, at trial $t$, given the current solution $\mathbf{w}_t$ and received loss functions $f_t^i(\mathbf{w}), i = 0, 1, \ldots, m$, we first estimate the constraint functions as

$$\widehat{f}_t^i(\mathbf{w}) = \frac{1}{t} \sum_{k=1}^{t} f_k^i(\mathbf{w}), i \in [m],$$

and then update the solution by $\mathbf{w}_{t+1} = \Pi_{\mathcal{W}_t}\left(\mathbf{w}_t - \eta \nabla f_t^0(\mathbf{w}_t)\right\}$ where $\eta > 0$ is the step size, $\Pi_{\mathcal{W}}(\mathbf{w}) = \min_{\mathbf{z} \in \mathcal{W}} \|\mathbf{z} - \mathbf{w}\|$ projects a solution $\mathbf{w}$ into domain $\mathcal{W}$, and $\mathcal{W}_t$ is an approximate domain given by $\mathcal{W}_t = \{\mathbf{w} : \widehat{f}_t^i(\mathbf{w}) \leq \gamma_i, i \in [m]\}$.

One problem with the above approach is that although it is feasible to satisfy all the constraints based on the true expected constraint functions, there is no guarantee that the approximate domain $\mathcal{W}_t$ is not empty. One way to address this issue is to estimate the expected constraint functions by burning the first $bT$ trials, where $b \in (0, 1)$ is a constant that needs to be adjusted to obtain the optimal performance, and keep the estimated constraint functions unchanged afterwards. Given the sampled functions $f_1^i, \ldots, f_{bT}^i$ received in the first $bT$ trials, we compute the approximate domain $\mathcal{W}'$ as

$$\widehat{f}^i(\mathbf{w}) = \frac{1}{bT} \sum_{t=1}^{bT} f_t^i(\mathbf{w}), i \in [m], \ \mathcal{W}' = \left\{\mathbf{w} : \widehat{f}^i(\mathbf{w}) \leq \gamma_i + \hat{\gamma}_i, i = 1, \ldots, m\right\}$$

where $\hat{\gamma}_i > 0$ is a relaxed constant introduced to ensure that with a high probability, the approximate domain $\mathcal{W}_t$ is not empty provided that the original domain $\mathcal{W}$ is not empty.

To ensure the correctness of the above approach, we need to establish some kind of uniform (strong) convergence assumption to make sure that the solutions obtained by projection onto the estimated domain $\mathcal{W}'$ will be close to the true domain $\mathcal{W}$ with high probability. It turns out that the following assumption ensures the desired property.

**Assumption 1** (Uniform Convergence). *Let $\widehat{f}^i(\mathbf{w}), i = 0, 1, \cdots, m$ be the estimated functions obtained by averaging over $bT$ i.i.d samples for $\bar{f}^i(\mathbf{w}), i \in [m]$. We assume that, with a high probability,*

$$\sup_{\mathbf{w} \in \mathcal{W}} \left|\widehat{f}^i(\mathbf{w}) - \bar{f}^i(\mathbf{w})\right| \leq O([bT]^{-q}), i = 0, 1, \cdots, m.$$

*where $q > 0$ decides the convergence rate.*

It is straightforward to show that under Assumption 1, with a high probability, for any $\mathbf{w} \in \mathcal{W}$, we have $\mathbf{w} \in \mathcal{W}'$, with appropriately chosen relaxation constant $\hat{\gamma}_i, i \in [m]$. Using the estimated domain $\mathcal{W}'$, for trial $t \in [bT + 1, T]$, we update the solution by $\mathbf{w}_{t+1} = \Pi_{\mathcal{W}'}(\mathbf{w}_t - \eta \nabla f_t^0(\mathbf{w}_t))$.

There are however several drawbacks with this naive approach. Since the first $bT$ trials are used for estimating the constraint functions, only the last $(1-b)T$ trials are used for searching for the optimal solution. The total amount of violation of individual constraint functions for the last $(1-b)T$ trials, given by $\sum_{t=bT+1}^{T} \bar{f}^i(\mathbf{w}_t)$, is $O((1-b)b^{-q}T^{1-q})$, where each of the $(1-b)T$ trials receives a violation of $O([bT]^{-q})$. Similarly, following the conventional analysis of online learning [26], we have $\sum_{t=bT+1}^{T}(f_t^0(\mathbf{w}_t) - f_t^0(\mathbf{w}_*)) \leq O(\sqrt{(1-b)T})$. Using the same trick as in [13], to obtain a solution with zero violation of constraints, we will have a regret bound $O((1-b)b^{-q}T^{1-q} + \sqrt{(1-b)T})$, which yields a convergence rate of $O(T^{-1/2} + T^{-q})$ which could be worse than the optimal rate $O(T^{-1/2})$ when $q < 1/2$. Additionally, this approach requires memorizing the constraint functions of the first $bT$ trials. This is in contrast to the typical assumption of online learning where only the solution is memorized.

**Remark 1.** *We finally remark on the uniform convergence assumption, which holds when the constraint functions are linear [25], but unfortunately does not hold for general convex Lipschitz functions. In particular, one can simply show examples where there is no uniform convergence for stochastic convex Lipchitz functions in infinite dimensional spaces [21]. Without uniform convergence assumption, the approximate domain $\mathcal{W}'$ may depart from the true $\mathcal{W}$ significantly at some unknown point, which makes the above approach to fail for general convex objectives.*

To address these limitations and in particular the dependence on uniform convergence assumption, we present an algorithm that does not require projection when updating the solution and does not require to impose any additional assumption on the stochastic functions except for the standard Lipschitz continuity assumption. We note that our result is closely related to the recent studies of learning from the viewpoint of optimization [23], which state that solutions found by stochastic gradient descent can be statistically consistent even when uniform convergence theorem does not hold.

---
**Algorithm 1** Stochastic Primal-Dual Optimization with Multiple Objectives
---
1: **INPUT**: step size $\eta$, $\boldsymbol{\lambda}_0 = (\lambda_0^1, \cdots, \lambda_0^m)$, $\lambda_0^i > 0, i \in [m]$ and total iterations $T$
2: $\mathbf{w}_1 = \boldsymbol{\lambda}_1 = \mathbf{0}$
3: **for** $t = 1, \ldots, T$ **do**
4:     Submit the solution $\mathbf{w}_t$
5:     Receive loss functions $f_t^i, i = 0, 1, \ldots, m$
6:     Compute the gradients $\nabla f_t^i(\mathbf{w}_t), i = 0, 1, \ldots, m$
7:     Update the solution $\mathbf{w}$ and $\boldsymbol{\lambda}$ by

$$\mathbf{w}_{t+1} = \Pi_{\mathcal{B}}\left(\mathbf{w}_t - \eta \nabla_{\mathbf{w}} \mathcal{L}_t(\mathbf{w}_t, \boldsymbol{\lambda}_t)\right) = \Pi_{\mathcal{B}}\left(\mathbf{w}_t - \eta \left[\nabla f_t^0(\mathbf{w}_t) + \sum_{i=1}^m \lambda_t^i \nabla f_t^i(\mathbf{w}_t)\right]\right),$$

$$\lambda_{t+1}^i = \Pi_{[0,\lambda_0^i]}\left(\lambda_t^i + \eta \nabla_{\lambda^i} \mathcal{L}_t(\mathbf{w}_t, \boldsymbol{\lambda}_t)\right) = \Pi_{[0,\lambda_0^i]}\left(\lambda_t^i + \eta \left[f_t^i(\mathbf{w}_t) - \gamma_i\right]\right).$$

8: **end for**
9: Return $\hat{\mathbf{w}}_T = \sum_{t=1}^T \mathbf{w}_t / T$
---

## 4 An Efficient Stochastic Primal-Dual Algorithm

We now turn to devise a tractable formulation of the problem, followed by an efficient primal-dual optimization algorithm and the statements of our main results. We show that with a high probability, the solution found by the proposed algorithm will exactly satisfy the expected constraints and achieves a regret bound of $O(\sqrt{T})$. The main idea of the proposed algorithm is to design an appropriate objective that combines the loss function $\bar{f}^0(\mathbf{w})$ with $\bar{f}^i(\mathbf{w}), i \in [m]$. As mentioned before, owing to the presence of conflicting goals and the randomness nature of the objective functions, we resort to seek for a solution that satisfies all the objectives instead of an optimal one. To this end, we define the following objective function

$$\bar{\mathcal{L}}(\mathbf{w}, \boldsymbol{\lambda}) = \bar{f}^0(\mathbf{w}) + \sum_{i=1}^m \lambda^i (\bar{f}^i(\mathbf{w}) - \gamma_i).$$

Note that the objective function consists of both the primal variable $\mathbf{w} \in \mathcal{W}$ and dual variable $\boldsymbol{\lambda} = (\lambda^1, \ldots, \lambda^m)^\top \in \Lambda$, where $\Lambda \subseteq \mathbb{R}_+^m$ is a compact convex set that bounds the set of dual variables and will be discussed later. In the proposed algorithm, we will simultaneously update solutions for both $\mathbf{w}$ and $\boldsymbol{\lambda}$. By exploring convex-concave optimization theory [16], we will show that with a high probability, the solution of regret $O(\sqrt{T})$ exactly obeyes the constraints.

As the first step, we consider a simple scenario where the obtained solution is allowed to violate the constraints. The detailed steps of our primal-dual algorithm is presented in Algorithm 1 . It follows the same procedure as convex-concave optimization. Since at each iteration, we only observed a randomly sampled loss functions $f_t^i(\mathbf{w}), i = 0, 1, \ldots, m$, the objective function given by

$$\mathcal{L}_t(\mathbf{w}, \boldsymbol{\lambda}) = f_t^0(\mathbf{w}) + \sum_{i=1}^m \lambda^i (f_t^i(\mathbf{w}) - \gamma_i)$$

provides an unbiased estimate of $\bar{\mathcal{L}}(\mathbf{w}, \boldsymbol{\lambda})$. Given the approximate objective $\mathcal{L}_t(\mathbf{w}, \boldsymbol{\lambda})$, the proposed algorithm tries to minimize the objective $\mathcal{L}_t(\mathbf{w}, \boldsymbol{\lambda})$ with respect to the primal variable $\mathbf{w}$ and maximize the objective with respect to the dual variable $\boldsymbol{\lambda}$.

To facilitate the analysis, we first rewrite the the constrained optimization problem

$$\min_{\mathbf{w} \in \mathcal{B} \cap \mathcal{W}} \bar{f}^0(\mathbf{w})$$

where $\mathcal{W}$ is defined as $\mathcal{W} = \left\{\mathbf{w} : \bar{f}^i(\mathbf{w}) \leq \gamma_i, i = 1, \ldots m\right\}$ in the following equivalent form:

$$\min_{\mathbf{w} \in \mathcal{B}} \max_{\boldsymbol{\lambda} \in \mathbb{R}_+^m} \bar{f}^0(\mathbf{w}) + \sum_{i=1}^m \lambda^i (\bar{f}^i(\mathbf{w}) - \gamma_i). \tag{2}$$

We denote by $\mathbf{w}_*$ and $\boldsymbol{\lambda}_* = (\lambda_*^1, \ldots, \lambda_*^m)^\top$ as the optimal primal and dual solutions to the above convex-concave optimization problem, respectively, i.e.,

$$\mathbf{w}_* = \arg\min_{\mathbf{w} \in \mathcal{B}} \bar{f}^0(\mathbf{w}) + \sum_{i=1}^m \lambda_*^i (\bar{f}^i(\mathbf{w}) - \gamma_i), \tag{3}$$

$$\boldsymbol{\lambda}_* = \arg\max_{\boldsymbol{\lambda} \in \mathbb{R}_+^m} \bar{f}^0(\mathbf{w}_*) + \sum_{i=1}^m \lambda^i (\bar{f}^i(\mathbf{w}_*) - \gamma_i). \tag{4}$$

The following assumption establishes upper bound on the gradients of $\mathcal{L}(\mathbf{w}, \boldsymbol{\lambda})$ with respect to $\mathbf{w}$ and $\boldsymbol{\lambda}$. We later show that this assumption holds under a mild condition on the objective functions.

**Assumption 2** (Gradient Boundedness). *The gradients $\nabla_{\mathbf{w}}\mathcal{L}(\mathbf{w}, \boldsymbol{\lambda})$ and $\nabla_{\boldsymbol{\lambda}}\mathcal{L}(\mathbf{w}, \boldsymbol{\lambda})$ are uniformly bounded, i.e., there exist a constant $G > 0$ such that*

$$\max\left(\nabla_{\mathbf{w}}\mathcal{L}(\mathbf{w}, \boldsymbol{\lambda}), \nabla_{\boldsymbol{\lambda}}\mathcal{L}(\mathbf{w}, \boldsymbol{\lambda})\right) \leq G, \ \ for\ any\ \ \mathbf{w} \in \mathcal{B} \ \ and \ \ \boldsymbol{\lambda} \in \Lambda.$$

Under the preceding assumption, in the following theorem, we show that under appropriate conditions, the average solution $\widehat{\mathbf{w}}_T$ generated by of Algorithm 1 attains a convergence rate of $O(1/\sqrt{T})$ for both the regret and the violation of the constraints.

**Theorem 1.** *Set $\lambda_0^i \geq \lambda_*^i + \theta, i \in [m]$, where $\theta > 0$ is a constant. Let $\widehat{\mathbf{w}}_T$ be the solution obtained by Algorithm 1 after $T$ iterations. Then, with a probability $1 - (2m+1)\delta$, we have*

$$\bar{f}^0(\widehat{\mathbf{w}}_T) - \bar{f}^0(\mathbf{w}_*) \leq \frac{\mu(\delta)}{\sqrt{T}} \ \ and \ \ \bar{f}^i(\widehat{\mathbf{w}}_T) - \gamma_i \leq \frac{\mu(\delta)}{\theta\sqrt{T}}, i \in [m]$$

*where $D^2 = \sum_{i=1}^m [\lambda_0^i]^2$, $\eta = [\sqrt{(R^2 + D^2)/2T}]/G$, and*

$$\mu(\delta) = \sqrt{2}G\sqrt{R^2 + D^2} + 2G(R + D)\sqrt{2\ln\frac{1}{\delta}}. \tag{5}$$

**Remark 2.** *The parameter $\theta \in \mathbb{R}_+$ is a quantity that may be set to obtain sharper upper bound on the violation of constraints and may be chosen arbitrarily. In particular, a larger value for $\theta$ imposes larger penalty on the violation of the constraints and results in a smaller violation for the objectives.*

We also can develop an algorithm that allows the solution to exactly satisfy all the constraints. To this end, we define $\widehat{\gamma}_i = \gamma_i - \frac{\mu(\delta)}{\theta\sqrt{T}}$. We will run Algorithm 1 but with $\gamma_i$ replaced by $\widehat{\gamma}_i$. Let $G'$ denote the upper bound in Assumption 2 for $\nabla_{\boldsymbol{\lambda}}\mathcal{L}(\mathbf{w}, \boldsymbol{\lambda})$ with $\widehat{\gamma}_i$ is replaced by $\gamma_i, i \in [m]$. The following theorem shows the property of the obtained average solution $\widehat{\mathbf{w}}_T$.

**Theorem 2.** *Let $\widehat{\mathbf{w}}_T$ be the solution obtained by Algorithm 1 with $\gamma_i$ replaced by $\widehat{\gamma}_i$ and $\lambda_0^i = \lambda_*^i + \theta, i \in [m]$. Then, with a probability $1 - (2m+1)\delta$, we have*

$$\bar{f}^0(\widehat{\mathbf{w}}_T) - \bar{f}^0(\mathbf{w}_*) \leq \frac{(1 + \sum_{i=1}^m \lambda_0^i)\mu'(\delta)}{\sqrt{T}} \ \ and \ \ \bar{f}^i(\widehat{\mathbf{w}}_T) \leq \gamma_i, i \in [m],$$

*where $\mu'(\delta)$ is same as (5) with $G$ is replaced by $G'$ and $\eta = [\sqrt{(R^2 + D^2)/2T}]/G'$.*

### 4.1 Convergence Analysis

Here we provide the proofs of main theorems stated above. We start by proving Theorem 1 and then extend it to prove Theorem 2.

*Proof.* (of Theorem 1) Using the standard analysis of convex-concave optimization, from the convexity of $\bar{\mathcal{L}}(\mathbf{w}, \cdot)$ with respect to $\mathbf{w}$ and concavity of $\bar{\mathcal{L}}(\cdot, \boldsymbol{\lambda})$ with respect to $\boldsymbol{\lambda}$, for any $\mathbf{w} \in \mathcal{B}$ and $\lambda^i \in [0, \lambda_0^i], i \in [m]$, we have

$\bar{\mathcal{L}}(\mathbf{w}_t, \boldsymbol{\lambda}) - \bar{\mathcal{L}}(\mathbf{w}, \boldsymbol{\lambda}_t)$

$\leq \ \langle \mathbf{w}_t - \mathbf{w}, \nabla_{\mathbf{w}}\bar{\mathcal{L}}(\mathbf{w}_t, \boldsymbol{\lambda}_t)\rangle - \langle \boldsymbol{\lambda}_t - \boldsymbol{\lambda}, \nabla_{\boldsymbol{\lambda}}\bar{\mathcal{L}}(\mathbf{w}_t, \boldsymbol{\lambda}_t)\rangle$

$= \ \langle \mathbf{w}_t - \mathbf{w}, \nabla_{\mathbf{w}}\mathcal{L}_t(\mathbf{w}_t, \boldsymbol{\lambda}_t)\rangle - \langle \boldsymbol{\lambda}_t - \boldsymbol{\lambda}, \nabla_{\boldsymbol{\lambda}}\mathcal{L}_t(\mathbf{w}_t, \boldsymbol{\lambda}_t)\rangle$

$\quad + \langle \mathbf{w}_t - \mathbf{w}, \nabla_{\mathbf{w}}\bar{\mathcal{L}}(\mathbf{w}_t, \boldsymbol{\lambda}_t) - \nabla_{\mathbf{w}}\mathcal{L}_t(\mathbf{w}_t, \boldsymbol{\lambda}_t)\rangle - \langle \boldsymbol{\lambda}_t - \boldsymbol{\lambda}, \nabla_{\boldsymbol{\lambda}}\bar{\mathcal{L}}(\mathbf{w}_t, \boldsymbol{\lambda}_t) - \nabla_{\boldsymbol{\lambda}}\mathcal{L}_t(\mathbf{w}_t, \boldsymbol{\lambda}_t)\rangle$

$\leq \ \frac{\|\mathbf{w}_t - \mathbf{w}\|^2 - \|\mathbf{w}_{t+1} - \mathbf{w}\|^2}{2\eta} + \frac{\|\boldsymbol{\lambda}_t - \boldsymbol{\lambda}\|^2 - \|\boldsymbol{\lambda}_{t+1} - \boldsymbol{\lambda}\|^2}{2\eta}$

$\quad + \frac{\eta}{2}\left(\|\nabla_{\mathbf{w}}\mathcal{L}_t(\mathbf{w}_t, \boldsymbol{\lambda}_t)\|^2 + \|\nabla_{\boldsymbol{\lambda}}\mathcal{L}_t(\mathbf{w}_t, \boldsymbol{\lambda}_t)\|^2\right)$

$\quad + \langle \mathbf{w}_t - \mathbf{w}, \nabla_{\mathbf{w}}\bar{\mathcal{L}}(\mathbf{w}_t, \boldsymbol{\lambda}_t) - \nabla_{\mathbf{w}}\mathcal{L}_t(\mathbf{w}_t, \boldsymbol{\lambda}_t)\rangle - \langle \boldsymbol{\lambda}_t - \boldsymbol{\lambda}, \nabla_{\boldsymbol{\lambda}}\bar{\mathcal{L}}(\mathbf{w}_t, \boldsymbol{\lambda}_t) - \nabla_{\boldsymbol{\lambda}}\mathcal{L}_t(\mathbf{w}_t, \boldsymbol{\lambda}_t)\rangle,$

where in the first inequality we have added and subtracted the stochastic gradients used for updating the solutions, the last inequality follows from the updating rules for $\mathbf{w}_{t+1}$ and $\boldsymbol{\lambda}_{t+1}$ and non-expensiveness property of the orthogonal projection operation onto the convex domain.

By adding all the inequalities together, we get

$$\sum_{t=1}^{T} \bar{\mathcal{L}}(\mathbf{w}_t, \boldsymbol{\lambda}) - \bar{\mathcal{L}}(\mathbf{w}, \boldsymbol{\lambda}_t)$$

$$\leq \quad \frac{\|\mathbf{w} - \mathbf{w}_1\|^2 + \|\boldsymbol{\lambda} - \boldsymbol{\lambda}_1\|^2}{2\eta} + \frac{\eta}{2} \sum_{t=1}^{T} \|\nabla_{\mathbf{w}} \mathcal{L}_t(\mathbf{w}_t, \boldsymbol{\lambda}_t)\|^2 + \|\nabla_{\boldsymbol{\lambda}} \mathcal{L}_t(\mathbf{w}_t, \boldsymbol{\lambda}_t)\|^2$$

$$+ \quad \sum_{t=1}^{T} \left\langle \mathbf{w}_t - \mathbf{w}, \nabla_{\mathbf{w}} \bar{\mathcal{L}}(\mathbf{w}_t, \boldsymbol{\lambda}_t) - \nabla_{\mathbf{w}} \mathcal{L}_t(\mathbf{w}_t, \boldsymbol{\lambda}_t) \right\rangle - \left\langle \boldsymbol{\lambda}_t - \boldsymbol{\lambda}, \nabla_{\boldsymbol{\lambda}} \bar{\mathcal{L}}(\mathbf{w}_t, \boldsymbol{\lambda}_t) - \nabla_{\boldsymbol{\lambda}} \mathcal{L}_t(\mathbf{w}_t, \boldsymbol{\lambda}_t) \right\rangle$$

$$\leq \quad \frac{R^2 + D^2}{2\eta} + \eta G^2 T$$

$$+ \quad \sum_{t=1}^{T} \left\langle \mathbf{w}_t - \mathbf{w}, \nabla_{\mathbf{w}} \bar{\mathcal{L}}(\mathbf{w}_t, \boldsymbol{\lambda}_t) - \nabla_{\mathbf{w}} \mathcal{L}_t(\mathbf{w}_t, \boldsymbol{\lambda}_t) \right\rangle - \left\langle \boldsymbol{\lambda}_t - \boldsymbol{\lambda}, \nabla_{\boldsymbol{\lambda}} \bar{\mathcal{L}}(\mathbf{w}_t, \boldsymbol{\lambda}_t) - \nabla_{\boldsymbol{\lambda}} \mathcal{L}_t(\mathbf{w}_t, \boldsymbol{\lambda}_t) \right\rangle$$

$$\leq \quad \frac{R^2 + D^2}{2\eta} + \eta G^2 T + 2G(R + D)\sqrt{2T \ln \frac{1}{\delta}} \quad \text{(w.p. } 1 - \delta\text{)},$$

where the last inequality follows from the Hoeffding inequality for Martingales [6]. By expanding the left hand side, substituting the stated value of $\eta$, and applying the Jensen's inequality for the average solutions $\widehat{\mathbf{w}}_T = \sum_{t=1}^{T} \mathbf{w}_t / T$ and $\widehat{\boldsymbol{\lambda}}_T = \sum_{t=1}^{T} \boldsymbol{\lambda}_t / T$, for any fixed $\lambda^i \in [0, \lambda_0^i], i \in [m]$ and $\mathbf{w} \in \mathcal{B}$, with a probability $1 - \delta$, we have

$$\bar{f}^0(\widehat{\mathbf{w}}_T) + \sum_{i=1}^{m} \lambda^i (\bar{f}^i(\widehat{\mathbf{w}}_T) - \gamma_i) - \bar{f}^0(\mathbf{w}) - \sum_{i=1}^{m} \widehat{\lambda}_T^i (\bar{f}^i(\mathbf{w}) - \gamma_i) \tag{6}$$

$$\leq \sqrt{2} G \sqrt{\frac{R^2 + D^2}{T}} + 2G(R + D)\sqrt{\frac{2}{T} \ln \frac{1}{\delta}}.$$

By fixing $\mathbf{w} = \mathbf{w}_*$ and $\boldsymbol{\lambda} = \mathbf{0}$ in (6), we have $\bar{f}^i(\mathbf{w}_*) \leq \gamma_i, i \in [m]$, and therefore, with a probability $1 - \delta$, have

$$\bar{f}^0(\widehat{\mathbf{w}}_T) \leq \bar{f}^0(\mathbf{w}_*) + \sqrt{2} G \sqrt{\frac{R^2 + D^2}{T}} + 2G(R + D)\sqrt{\frac{2}{T} \ln \frac{1}{\delta}}.$$

To bound the violation of constraints we set $\mathbf{w} = \mathbf{w}_*$, $\lambda^i = \lambda_0^i, i \in [m]$, and $\lambda^j = \lambda_*^j, j \neq i$ in (6). We have

$$\bar{f}^0(\widehat{\mathbf{w}}_T) + \lambda_0^i (\bar{f}^i(\widehat{\mathbf{w}}_T) - \gamma_i) + \sum_{j \neq i} \lambda_*^j (\bar{f}^j(\widehat{\mathbf{w}}_T) - \gamma_j) - \bar{f}^0(\mathbf{w}_*) - \sum_{i=1}^{m} \widehat{\lambda}_T^i (\bar{f}^i(\mathbf{w}_*) - \gamma_i)$$

$$\geq \quad \bar{f}^0(\widehat{\mathbf{w}}_T) + \lambda_0^i (\bar{f}^i(\widehat{\mathbf{w}}_T) - \gamma_i) + \sum_{j \neq i} \lambda_*^j (\bar{f}^j(\widehat{\mathbf{w}}_T) - \gamma_j) - \bar{f}^0(\mathbf{w}_*) - \sum_{i=1}^{m} \lambda_*^i (\bar{f}^i(\mathbf{w}_*) - \gamma_i)$$

$$\geq \quad \theta(\bar{f}^i(\widehat{\mathbf{w}}_T) - \gamma_i),$$

where the first inequality utilizes (4) and the second inequality utilizes (3). We thus have, with a probability $1 - \delta$,

$$\bar{f}^i(\widehat{\mathbf{w}}_T) - \gamma_i \leq \frac{\sqrt{2} G}{\theta} \sqrt{\frac{R^2 + D^2}{T}} + \frac{2G(R + D)}{\theta} \sqrt{\frac{2}{T} \ln \frac{1}{\delta}}, i \in [m].$$

We complete the proof by taking the union bound over all the random events. $\qquad \square$

We now turn to the proof of Theorem 2 that gives high probability bound on the convergence of the modified algorithm which obeys all the constraints.

*Proof.* (of Theorem 2) Following the proof of Theorem 1, with a probability $1 - \delta$, we have

$$\bar{f}^0(\widehat{\mathbf{w}}_T) + \sum_{i=1}^{m} \lambda^i (\bar{f}^i(\widehat{\mathbf{w}}_T) - \widehat{\gamma}_i) - \bar{f}^0(\mathbf{w}) - \sum_{i=1}^{m} \widehat{\lambda}_T^i (\bar{f}^i(\mathbf{w}) - \widehat{\gamma}_i)$$

$$\leq \sqrt{2} G' \sqrt{\frac{R^2 + D^2}{T}} + 2G'(R + D)\sqrt{\frac{2}{T} \ln \frac{1}{\delta}}$$

Define $\widetilde{\mathbf{w}}_*$ and $\widetilde{\boldsymbol{\lambda}}_*$ be the saddle point for the following minimax optimization problem

$$\min_{\mathbf{w}\in\mathcal{B}} \max_{\boldsymbol{\lambda}\in\mathbb{R}^m_+} \bar{f}^0(\mathbf{w}) + \sum_{i=1}^m \lambda^i(\bar{f}^i(\mathbf{w}) - \widehat{\gamma}_i)$$

Following the same analysis as Theorem 1, for each $i \in [m]$, by setting $\mathbf{w} = \widetilde{\mathbf{w}}_*$, $\lambda^i = \lambda^i_0$, and $\lambda^j = \widetilde{\lambda}^j_*$, using the fact that $\widetilde{\lambda}^j_* \leq \lambda^j_*$, we have, with a probability $1 - \delta$

$$\theta(\bar{f}^i(\widehat{\mathbf{w}}_T) - \gamma_i) \leq \sqrt{2}G'\sqrt{\frac{R^2 + D^2}{T}} + 2G'(R + D)\sqrt{\frac{2}{T}\ln\frac{1}{\delta}} - \frac{\mu(\delta)}{\sqrt{T}} \leq 0,$$

which completes the proof. $\qquad\square$

## 4.2 Implementation Issues

In order to run Algorithm 1, we need to estimate the parameter $\lambda^i_0, i \in [m]$, which requires to decide the set $\Lambda$ by estimating an upper bound for the optimal dual variables $\lambda^i_*, i \in [m]$. To this end, we consider an alternative problem to the convex-concave optimization problem in (2), i.e.

$$\min_{\mathbf{w}\in\mathcal{B}} \max_{\lambda\geq 0} \bar{f}^0(\mathbf{w}) + \lambda \max_{1\leq i\leq m} (\bar{f}^i(\mathbf{w}) - \gamma_i). \tag{7}$$

Evidently $\mathbf{w}_*$ is the optimal primal solution to (7). Let $\lambda_a$ be the optimal dual solution to the problem in (7). We have the following proposition that links $\lambda^i_*, i \in [m]$, the optimal dual solution to (2), with $\lambda_a$, the optimal dual solution to (7).

**Proposition 1.** *Let $\lambda_a$ be the optimal dual solution to (7) and $\lambda^i_*, i \in [m]$ be the optimal solution to (2). We have $\lambda_a = \sum_{i=1}^m \lambda^i_*$.*

*Proof.* We can rewrite (7) as $\min_{\mathbf{w}\in\mathcal{B}} \max_{\lambda\geq 0, \mathbf{p}\in\Delta_m} \bar{f}^0(\mathbf{w}) + \sum_{i=1}^m p_i\lambda(\bar{f}^i(\mathbf{w}) - \gamma_i)$, where domain $\Delta_m$ is defined as $\Delta_m = \{\alpha \in \mathbb{R}^m_+ : \sum_{i=1}^m \alpha_i = 1\}$. By redefining $\lambda^i = p_i\lambda$, we have the problem in (7) equivalent to (2) and consequently $\lambda = \sum_{i=1}^m \lambda^i$ as claimed. $\qquad\square$

Given the result from Proposition 1, it is sufficient to bound $\lambda_a$. In order to bound $\lambda_a$, we need to make certain assumption about $\bar{f}^i(\mathbf{w}), i \in [m]$. The purpose of introducing this assumption is to ensure that the optimal dual variable is well bounded from the above.

**Assumption 3.** *We assume $\min_{\alpha\in\Delta_m} \left\|\sum_{i=1}^m \alpha_i\nabla\bar{f}^i(\mathbf{w})\right\| \geq \tau$, where $\tau > 0$ is a constant.*

Equipped with Assumption 3, we are able to bound $\lambda_a$ by $\tau$. To this end, using the first order optimality condition of (2) [7], we have $\lambda_a = \|\nabla\bar{f}^0(\mathbf{w}_*)\|/\|\partial g(\mathbf{w})\|$, where $g(\mathbf{w}) = \max_{1\leq i\leq m} \bar{f}^i(\mathbf{w})$. Since $\partial g(\mathbf{w}) \in \left\{\sum_{i=1}^m \alpha_i\nabla\bar{f}^i(\mathbf{w}) : \alpha \in \Delta_m\right\}$, under Assumption 3, we have $\lambda_a \leq \frac{L}{\tau}$. By combining Proposition 1 with the upper bound on $\lambda_a$, we obtain $\lambda^i_* \leq \frac{L}{\tau}, i \in [m]$ as desired.

Finally, we note that by having $\boldsymbol{\lambda}_*$ bounded, Assumption 2 is guaranteed by setting $G^2 = \max(L^2\left(1 + \sum_{i=1}^m \lambda^i_0\right)^2, \max_{\mathbf{w}\in\mathcal{B}} \sum_{i=1}^m (\bar{f}^i(\mathbf{w}) - \gamma_i)^2)$ which follows from Lipschitz continuity of the objective functions. In a similar way we can set $G'$ in Theorem 2 by replacing $\gamma_i$ with $\widehat{\gamma}_i$.

## 5 Conclusions and Open Questions

In this paper we have addressed the problem of stochastic convex optimization with multiple objectives underlying many applications in machine learning. We first examined a simple problem reduction technique that eliminates the stochastic aspect of constraint functions by approximating them using the sampled functions from each iteration. We showed that this simple idea fails to attain the optimal convergence rate and requires to impose a strong assumption, i.e., uniform convergence, on the objective functions. Then, we presented a novel efficient primal-dual algorithm which attains the optimal convergence rate $O(1/\sqrt{T})$ for all the objectives relying only on the Lipschitz continuity of the objective functions. This work leaves few direction for further elaboration. In particular, it would be interesting to see whether or not making stronger assumptions on the analytical properties of objective functions such as smoothness or strong convexity may yield improved convergence rate.

**Acknowledgments.** The authors would like to thank the anonymous reviewers for their helpful and insightful comments. The work of M. Mahdavi and R. Jin was supported in part by ONR Award N000141210431 and NSF (IIS-1251031).

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
