[Reviews · NeurIPS 2013]

Submitted by Assigned_Reviewer_1

This paper proposes an approach to stochastic multi-objective optimization. The main idea is simply described: optimize a single objective while taking other objectives as constraints. The authors proposes a primal-dual stochastic optimization algorithm to solve the problem and prove that it achieves (for the primal objective) the optimal 1/\sqrt{T} convergence rate.

As far as I am concerned, the theory is solid and it does provide a good insight into the problem of interest. The authors didn't mention how to set the parameters gamma_i, which may be difficult for specific problem. In fact, the challenge of setting gamma_i is not necessarily lower than the difficulty of setting mixture weights in the linear scalarization approach. The paper could be more interesting if we don't need gamma_i in advance.

It is nice to see lambda^0's can be upper bounded in section 4.2. It would be interesting to discuss the optimal choice of \theta in certain circumstances. The paper is well-written and the results (as far as I know) are new. It is nice to see that the stochastic method well fit the multi-objective problem even though the choice of gamma_i is still a problem.
Summary: A good paper overall. Interesting theoretical results with rigorous arguments. Would be better if including some empirical evaluation.

Submitted by Assigned_Reviewer_4

This paper presents an algorithm for multiple objectives stochastic convex optimization.
The issue with multiple objectives is handled by choosing one objective to minimize and by
constraining all the others via lower bounds. The stochastic sampling is addressed by averaging
the costs on the available samples. The proposed algorithm is then a primal-dual projected gradient descent.
The authors show also that 2 naive implementations yield a worse solution error bound than the primal-dual
approach.

Quality
--
The paper seems technically sound. I have checked several steps but not all.
The main serious shortcoming is that there are no experiments at all to support
any of the claims made in the paper.
Overall, this seems an unfinished paper. Although this is mostly a theoretical
paper, I'd recommend to add experiments to show advantages and limitations
of the proposed strategy (transfer m objectives to constraints).
For example, how does one choose the bounds for each objective function?
What is the best way to pick the objective to be minimized?
Can one draw any connection of Pareto optimal solutions?

Clarity
--
Overall the paper is clear.
The authors should also show some illustrations together with the theory.
Typos/unclear
line 183: what is \hat f_i ?
line 215: to fail for -> fail for


Originality
--
The originality of this work is difficult to assess.
It seems that Algorithm 1 and the formal proofs are new,
but I am not an expert in the field and relevant literature
could have escaped me. The idea of moving all but one
objectives to the set of constraints seems quite simple
and with several limitations.


Significance
--
The use of multiple objectives as well as stochastic sampling
is present in a few applications (some shown at the end of
the Introduction) and worthy of attention.
However, the resulting algorithm is quite a straightforward
idea, and has some limitations (optimization is with respect to only
one objective function) that are not fully examined in the paper.
The full significance of the algorithm is not demonstrated with
experiments either. The main contribution then is left
to the proof of the error bounds (Theorems 1 and 2).

Summary: Multiobjective optimization can be tackled by selecting one cost and
imposing bounds on the other costs. There are no experiments to
validate all the theory.

Submitted by Assigned_Reviewer_5

This paper considered the setting where we want to optimize one function \bar{f]_0 when there are constraints on other functions \bar{f}_1...\bar{f}f_m, but our access is only stochastic for both the objective and the constraints. Naive approaches don't work, but a very natural primal-dual algorithm works to get a rate of 1/\sqrt T.

Overall, I was happy with almost all aspects of the paper - however the convergence analysis can be made clearer and more intuitive. While it was shown that simple things don't work, the algorithm that does work is quite simple and probably one of the first ones that one would think of - this could be a pro (someone has to prove it, and it might not be obvious how to) or a con (the result is not that surprising). However, it is a complete work, most of my concerns were addressed, it was fairly easy to follow, clear and original.

Minor typos :
At many points there is \lambda^*_i and elsewhere \lambda^i_*. I assumed this was a typo and that you refer to the same object everywhere.
Summary: I recommend this paper for acceptance, because it is clear and fairly original, of high quality, and of significance to the community, but the surprise factor is low.
Author Feedback

Author rebuttal: We are grateful to all anonymous reviewers for their useful and constructive comments and PC members for handling our paper.

========= Reviewer 1 ==========
Q: Setting gamma_i is still challenging.

A: We agree that \gamma_i needs to be set appropriately, but it is eminently worthy to emphasize that setting \gamma_i would be relatively easier than setting the combination weights required in the scalarization method. For instance, in maximizing both precision and recall, Neyman–Pearson classification, and the finance example we have discussed at the beginning of the article, this can be done by some prior knowledge about the problem's domain (e.g., [1] and [2] discuss this for few applications). Also, as discussed, the proposed primal-dual algorithm guarantees bound on *ALL* the objectives, while solving the weighted combination of objectives does not necessarily guarantee any bound on single objectives which is the main goal of the proposed method.

[1] Clayton Scott, Performance Measures for Neyman–Pearson Classification, IEEE-IT, 2007.
[2] Clayton Scott and Robert Nowak, A Neyman–Pearson Approach to Statistical Learning, IEEE-IT, 2005.

Q: It is nice to see lambda^0's can be upper bounded in Section 4.2. It would be interesting to discuss the optimal choice of \theta in certain circumstances.

A: Thanks a lot for pointing these facts out. We will resolve these issues and include discussion about these parameters in the setting of three problems we discussed in the Introduction.

Q: Would be better if including some empirical evaluation

A: Regarding the empirical evaluation of the proposed algorithms, we would like to mention that the application of the algorithm to Neyman–Pearson classification and finance problem will be included in an extended version. Also, in this paper we only considered simple Lipschitz continuous objectives and the generalization to the objective with stronger assumptions on the objectives such as smoothness and strong convexity to obtain better convergence rate will be discussed in an extended work as well.

========= Reviewer 2 ==========

Q: Although this is mostly a theoretical paper, I'd recommend to add experiments to show advantages and limitations of the proposed strategy.

A: We agree with the reviewer that having experimental results definitely makes this work more complete and we will include empirical studies along with the detailed algorithm for each specific application and generalization of the algorithm to smooth and strongly convex functions in an extended version of this work.

Q: Only optimizing one objective function with the rest added to the constraints, too simple, limited significance.

A: Indeed, this strategy is popular in the study of multi-objective optimization. But in this work we showed that in stochastic optimization, the proposed primal-dual algorithm is able to bound *ALL* the objectives. Solving the reduced problem with a standard stochastic optimization algorithm does not work as we discussed, even under strong assumptions on the stochastic objectives. Hence, our work departs from previous literature in our treatment of objectives and guarantee on the individual objectives.


========= Reviewer 3 ==========
Thanks a lot for pointing out the typo. We will fix that and do our best to make the convergence analysis more clear and intuitive.